# N-Doped Porous Carbon-Nanofiber-Supported Fe_3_C/Fe_2_O_3_ Nanoparticles as Anode for High-Performance Supercapacitors

**DOI:** 10.3390/molecules28155751

**Published:** 2023-07-30

**Authors:** Li Li, Fengting Xie, Heyu Wu, Yuanyuan Zhu, Pinghua Zhang, Yanjiang Li, Hengzheng Li, Litao Zhao, Guang Zhu

**Affiliations:** 1Key Laboratory of Spin Electron and Nanomaterials of Anhui Higher Education Institutes, Suzhou University, Suzhou 234000, China; lili@ahszu.edu.cn (L.L.); coco9024137@163.com (F.X.); wuheyu19991105@163.com (H.W.); zhuyy@dicp.ac.cn (Y.Z.); yjli@ahszu.edu.cn (Y.L.); lihengzheng@ahszu.edu.cn (H.L.); ltzhao@ahszu.edu.cn (L.Z.); 2State Key Laboratory of Catalysis, Dalian Institute of Chemical Physics Chinese Academy of Sciences, Dalian 116023, China; 3School of Chemistry and Materials Science, University of Science and Technology of China, Hefei 230026, China; 4Institute for Carbon Neutralization, College of Chemistry and Materials Engineering, Wenzhou University, Wenzhou 325035, China

**Keywords:** Fe_3_C/Fe_2_O_3_, electrospinning, N-doped carbon nanofibers, supercapacitor

## Abstract

Exploring anode materials with an excellent electrochemical performance is of great significance for supercapacitor applications. In this work, a N-doped-carbon-nanofiber (NCNF)-supported Fe_3_C/Fe_2_O_3_ nanoparticle (NCFCO) composite was synthesized via the facile carbonizing and subsequent annealing of electrospinning nanofibers containing an Fe source. In the hybrid structure, the porous carbon nanofibers used as a substrate could provide fast electron and ion transport for the Faradic reactions of Fe_3_C/Fe_2_O_3_ during charge–discharge cycling. The as-obtained NCFCO yields a high specific capacitance of 590.1 F g^−1^ at 2 A g^−1^, superior to that of NCNF-supported Fe_3_C nanoparticles (NCFC, 261.7 F g^−1^), and NCNFs/Fe_2_O_3_ (NCFO, 398.3 F g^−1^). The asymmetric supercapacitor, which was assembled using the NCFCO anode and activated carbon cathode, delivered a large energy density of 14.2 Wh kg^−1^ at 800 W kg^−1^. Additionally, it demonstrated an impressive capacitance retention of 96.7%, even after 10,000 cycles. The superior electrochemical performance can be ascribed to the synergistic contributions of NCNF and Fe_3_C/Fe_2_O_3_.

## 1. Introduction

The development and application of renewable energy resources such as solar, wind, and water energy have created an urgent need for high-efficiency electric power sources in recent decades [1,2]. Among them, supercapacitors (SCs) have received tremendous attention, owing to their unique features such as a low maintenance cost, environmental friendliness, and a high charge efficiency, as well as a long cyclic life [3,4]. Nevertheless, SCs always suffering from a low energy density, which leads to poor endurance, and seriously hampers large-scale practical applications. According to the formula of energy density (*E =* 0.5*CV*^2^) of SCs [5,6], the primary strategies for obtaining high-performance devices are to utilize high-specific-capacitance (*C*) active materials that store charges relying on the Faradic reactions and/or electrical double layer capacitance (EDLC), and to design an asymmetric structure to extend the operation voltage window (*V*) [7,8].

To date, massive efforts have focused on exploring electrode materials with a high capacitive performance and long-term cycling stability [9,10]. In prior works, diverse active materials have been applied to construct electrodes for high-performance SC applications, such as transition metal oxides/carbides/sulfides/phosphides, carbonaceous materials, conducting polymers [11,12], etc. Among them, the Fe-based oxides/carbides (e.g., Fe_2_O_3_, Fe_3_O_4_, Fe_3_C, etc.) exhibit an outstanding electrochemical performance, with aspects such as a high theoretical specific capacitance, a large voltage window, and multivalent states [13,14]. Moreover, the multiple crystalline forms, low toxicity, rich resources, and low cost impel them to be extensively used as an anode for SCs [15]. Nevertheless, they also suffer from poor conductivity and a disappointing structural stability during the electrochemical reaction process, leading to a bad rate capacity and poor cyclic life [16,17,18]. To address these issues, carbonaceous materials (e.g., carbon cloth, carbon nanoarray, graphene, etc.) are widely used in combining Fe-based oxides/carbides, to design a high-performance electrode material relying on their excellent conductivity, outstanding stability, and environmental friendliness [19,20]. For example, Yuan et al. prepared ultrasmall Fe_2_O_3_-embedded carbon nanotubes through facile dipping and combustion processes, using carbon cloth as the substrate [21]. Benefiting from its unique architecture, the optimized sample displayed outstanding hydrophilia and delivered the high specific capacitance of 483.4 mF cm^−2^. The flexible device constructed using this nanostructure composite exhibited the high working voltage of 2.0 V, and delivered the large energy density of 0.11 mWh cm^−2^. To further enhance the electrochemical performance of an Fe-based oxide/porous carbon composite, heteroatom-doped porous carbon nanomaterials are used to anchored Fe-based oxides/carbides. These heteroatoms (such as N, S, P, etc.) in carbon lattice are facilitated, to enhance electrical conductivity, and can further contribute a remarkable pseudocapacitance to the as-obtained porous carbon nanomaterials, which has a cooperative effect on the preparative high-performance electrode materials [22,23,24]. For instance, Xia et al. fabricated N-doped-carbon-nanoarray-supported Fe_2_O_3_ nanoneedles on carbon cloth, through immersion and subsequent hydrothermal steps [25]. This hybrid structure could enhance the conductivity of Fe_2_O_3_, and provide sufficient diffusion channels for Faradic reactions and ion transfer. A symmetric SC based on the as-obtained composite showed the high energy density of 14.1 Wh kg^−1^.

Although locating Fe-based oxides/carbides on carbonaceous substrates could effectively lead to the construction of high-performance electrode materials for SCs, the complex preparation procedures and high cost mean that it is still difficult for them to meet the demands of practical application. Additionally, the carbonaceous substrates, such as carbon cloth, with a poor capacitance contribution and large mass/volume proportion would lead to a low gravimetric/volumetric specific capacitance in the as-assembled SC devices [26], which is not conducive to the development of lightweight devices. Compared to other carbonaceous materials, electrospinning carbon nanofibers (ECNFs) with advantageous architecture, such as a large specific surface area (SSA), and functional integration characteristics are recognized as an ideal substrate and electrode material for SCs, and have been the subject of tremendous research in recent years [27,28,29]. For example, Yang et al. located Fe^3+^ onto electrospinning nanofiber precursors via ion exchanging [14]. After carbonizing at high temperatures, the Fe_x_C_y_/Fe nanoparticles were inserted into N-doped ECNFs. The as-obtained flexible electrode displayed superior electrochemical performances (340 F g^−1^ at 1 A g^−1^). Accordingly, anchoring Fe-based oxides/carbides into N-doped ECNFs is a valid way to construct excellent electrode materials for SCs. Nevertheless, the preparation methods (such as ion exchange, hydrothermal synthesis, and electrodeposition) of Fe-based oxides/carbides/N-doped ECNFs generally involve tedious procedures that cannot meet the demands of large-scale application. More importantly, the hybrid architecture composites fabricated through these methods show a loose contact between Fe-based oxides/carbides and ECNFs, which is adverse for electron transfer and for buffering the volume changes during the charge–discharge process, leading to a disappointing rate capability and significant deterioration of the cycling performance. Hence, researching facile synthesis methods to design Fe-based oxide/carbide/N-doped ECNF composites with an excellent electrochemical performance should be further attempted.

Herein, an Fe_3_C/Fe_2_O_3_-anchored N-doped ECNF (NCFCO) hybrid structure nanocomposite was prepared by facile-carbonizing and oxidizing the electrospun nanofiber precursor containing polyacrylonitrile and iron acetylacetonate, and its electrochemical performance as an electrode for SCs was assessed using different electrochemical measurements. Benefiting from the synergetic effect of the Fe_3_C/Fe_2_O_3_ and N-doped ECNFs, the as-prepared NCFCO exhibited a remarkably reinforced specific capacitance of 590.1 F g^−1^ at 2 A g^−1^, much higher than those of Fe_3_C-anchored N-doped ECNFs (NCFC, 261.7 F g^−1^) and Fe_2_O_3_ nanoparticle/N-doped ECNF composite (NCFO, 398.3 F g^−1^). Importantly, the asymmetric SC based on our NCFCO anode and commercial activated carbon (AC) cathode delivered the high energy density of 14.2 Wh kg^−1^ at 800 W kg^−1^, and the surface morphology of the NCFCO electrode showed almost no change after long-cycling, demonstrating that integrating Fe-based oxides/carbides into N-doped ECNFs by electrospinning, to construct high-performance electrode material for SC applications, is an extremely effective method.

## 2. Results and Discussion

The preparation procedures of the NCFCO are shown in Figure 1. As can be seen, the nanofiber precursor composed of PAN and IAA is prepared through electrospinning. During the subsequent carbonization procedure in a N_2_ atmosphere, the PAN degrades into porous the N-doped carbon nanofiber (NCNF) [30], while the Fe atoms gather together to form Fe_3_C nanoparticles, and are inserted into the carbon nanofiber skeletons, which is likely to be due to the thermal motion deriving molecule accumulation at a high temperature [31]. After being calcined in air, the porous NCNF substrate can be reserved, while the Fe_3_C nanoparticles transform into the Fe_3_C/Fe_2_O_3_ composite, owing to the inadequate oxidation at a relatively low temperature of 200 °C. Notably, the oxidization temperature significantly determines the composition of the as-prepared products. Excessive temperatures lead to a higher level of carbon component depletion, and the complete conversion of Fe_3_C to Fe_2_O_3_.

### 2.1. Characterizations

For ECNF (Figure 2a), the carbon nanofibers present smooth surfaces, and interconnect with each other to form a disorder network architecture. Figure 2b displays the scanning electron microscopic (SEM) image of the NCFC, which also shows a nanofiber morphology with many Fe_3_C nanoparticles anchoring on the surface of the carbon nanofibers, to form an Fe_3_C/N-doped ECNF hybrid architecture. Compared with the morphology of the precursor after electrospinning (Appendix A), the nanofiber profile of the NCFC is well-reserved during the carbonization process, and additionally, Fe_3_C nanoparticles are generated from the aggregating of Fe atoms. The NCFCO reveals a similar profile to that of the NCFC (Figure 2c), suggesting that the carbon skeleton has not been destroyed after oxidizing at 200 °C. Nevertheless, when the temperature rises to 400 °C, only a few carbon nanofibers can be observed, owing to the fact that most of the carbon components are consumed through reacting with oxygen in the air at high temperatures (Figure 2d). Simultaneously, the Fe_3_C nanoparticles are completely oxidized to Fe_2_O_3_. Figure 2e–i show the elemental mappings of the NCFCO. Obviously, the C, O, and N elements are distributed evenly on the surface of the NCFCO, while the Fe spectroscopy mainly originates from the position of the nanoparticles, demonstrating that the nanoparticles located on the surface of carbon nanofibers are generated by the gathering of Fe atoms. The existence of the C, O, N, and Fe elements can also be proved using the energy-dispersive X-ray spectroscopy of the NCFCO (Appendix A). It is worth mentioning that the hybrid architecture of Fe_3_C/Fe_2_O_3_/NCNFs is developed through slow atom diffusion at a high temperature, which could greatly facilitate electron transfer, and prevent the structure of Fe_2_O_3_/Fe_2_O_3_ from being damaged during long-term cycling, owing to the tight connection between the heterogenous components.

The crystal phases and compositions of the NCFC, NCFCO, and NCFO are identified by applying X-ray diffraction (XRD) patterns (Figure 3a) and Raman spectra (Figure 3b). It can be observed from Figure 3a that both the NCFC and NCFCO display obvious characteristic peaks at 37.9°, 39.8°, 40.7°, 43.0°, 43.9° 44.9°, 45.8°, 49.3°, and 70.8°, corresponding to the (210), (002), (201), (211), (102), (031), (112), (221), and (123) planes of Fe_3_C (JCPDS: no. 35-0772), respectively. Additionally, for the NCFCO and NCFO, apparent diffraction peaks at the angle of 24.1°, 33.1°, 35.6° 40.8°, 49.5°, 54.1°, 57.5°, 62.4°, and 64.0° can be observed, indexing to the (012), (104), (110), (113), (024), (116), (122), (214), and (300) planes of Fe_2_O_3_ (JCPDS: no. 33-0664). In particular, no Fe_3_C characteristic peaks can be detected, implying that all of the Fe_3_C has been oxidized to Fe_2_O_3_ in the NCFO. This can also be demonstrated using Raman spectroscopy. As revealed in Figure 3b, for the NCFC and NCFCO, two typical peaks, located at 1353 (D band) and 1590 cm^−1^ (G band), can be assigned to the disorder and sp^2^ hybridized carbon atoms in the carbon lattice, respectively [32,33]. Furthermore, the slight peaks at about 224, 294, and 393 cm^−1^ belong to the A_g1_, E_g2_ + E_g3_, and E_g4_ modes of Fe_2_O_3_, respectively [34]. The above results prove that the NCFCO consists of Fe_3_C, Fe_2_O_3_, and amorphous carbon. The Raman spectroscopy of the NCFO displays sharp peaks at around 224, 294, and 393 cm^−1^, demonstrating the existence of Fe_2_O_3_, which is in accordance with its XRD result.

The N_2_ adsorption–desorption isotherms (Figure 3c) were recorded, to investigate the porous character of the samples. As can be seen, all of the samples display type-IV isotherms, demonstrating the existence of mesopores in these hybrid architectures [35]. Additionally, the sharp uptake in the low relative pressure region suggests an abundant micropore content in the NCFC and NCFCO, and the steep slope in the high relative pressure zone is owing to the existence of macropores [36]. The pore-size distribution curves (Figure 3d) also prove the presence of micropores and mesopores in the samples. Therefore, both the NCFC and NCFCO are hierarchical pore architectures. Generally, micropores are the significant architecture to store ions, and mesopores can facilitate ion transfer and migration, whereas macropores are efficient buffer regions for electrolyte ions. Therefore, the hierarchically porous structure in the NCFCO is beneficial to realizing a superior electrochemical performance. The SSA, pore volume, and average pore diameter of the NCFC, NCFCO, and NCFO are summarized in Appendix A. Clearly, as the calcined temperature increases, the SSA and pore volume are all decreasing, which is likely due to the fact that more carbon components are drained at higher temperatures, thus leaving a larger average pore diameter.

The surface elemental valences and chemical compositions of the NCFC, NCFCO, and NCFO were researched by applying X-ray photoelectron spectroscopy (XPS). The XPS survey spectra of the samples are presented in Figure 4a. As can be clearly seen, there are C1s, O1s, and Fe2p peaks at around 286, 530, and 723 eV, respectively. Compared with the NCFO, both the NCFC and NCFCO exhibit apparent N1s peaks at the binding energy of about 400 eV, which is likely due to fewer NCNFs having been retained in the NCFO after oxidization at the higher temperature of 400 °C. The detailed elemental contents of the NCFC, NCFCO, and NCFO are listed in Table 1. Clearly, as the calcined temperature increases, more carbon components are consumed, while the Fe atoms are reserved in the form of Fe-based oxide/carbide. The high-resolution C1 spectra (Appendix A) of the NCFC, NCFCO, and NCFO can be deconvoluted into four peaks, centered at 284.2, 285.9, 286.9, and 288.7 eV, which can be assigned to the C–C/C=C, C–O, C–N/C=N, and O–C=O bonds [37,38], respectively. Additionally, for the NCFC, the peak at the binding energy of 283.5 eV can be clearly observed, corresponding with the Fe–C bond. The N1s spectra (Figure 4b and Appendix A) could be segmented into the pyridinic N (N–6), pyrrolic N (N–5), graphitic N (N–Q), and oxidic N (N–X), situated at 397.7, 398.4, 400.0, and 401.3 eV, respectively, indicating that the N atoms were successfully doped into the carbon lattice, with different bonding modes [39]. It is worth mentioning that the N–Q is beneficial in enhancing the conductivity of the carbon matrix, and therefore in improving its electrochemical performance [23,40]. Furthermore, the N–6 and N–5 species can contribute effective pseudocapacitance during the charge–discharge process, as well [29,41]. In the deconvoluted O1 spectra presented in Figure 4c and Appendix A, four peaks, located at 529.8, 531.1, 532.6, and 533.3 eV, can be ascribed to the Fe–O, C=O, C–O, and C–OOH species [13,36,42], respectively. Figure 4d and Appendix A show the high-resolution Fe2p spectra. The photoelectron peaks at 710.7 and 723.6 eV correspond to the Fe^2+^2p_3/2_ and Fe^2+^2p_1/2_ orbitals, while the peaks at 714.1 and 726.6 eV belong to the Fe^3+^2p_3/2_ and Fe^3+^2p_1/2_ orbitals [43], respectively. Meanwhile, two accompanying shake-up satellite peaks at 719.2 and 731.8 eV can also be observed [44].

### 2.2. Electrochemical Measurements

The electrochemical performances of the NCFC, NCFCO, and NCFO are detected using electrochemical impedance spectroscopy (EIS), cyclic voltammetry (CV), and galvanostatic charge–discharge (GCD) techniques in 2 mol L^−1^ KOH electrolyte, and the results are revealed in Figure 5. As presented, all of the EIS spectra (Figure 5a) display approximate straight lines in the low frequency region, indicating a superior capacitive property [45,46]. The charge transfer resistances (R_ct_) of the as-prepared samples are fitted by means of the equivalent circuit (inset of Figure 5a). As expected, the NCFCO exhibits the smallest R_ct_ (1.2 Ω), compared to those of the NCFC (15.7 Ω) and NCFO (36.3 Ω), indicating the fastest electrode reaction kinetics. Such a result can be attributed to its large SSA, enlarged pore volume, and the synergistic effects between the Fe_3_C/Fe_2_O_3_ and the residual carbon skeleton. Additionally, it can be found that the equivalent series resistances (R_s_) of the samples increase with the increasing oxidization temperature (in the inset of Figure 5a), owing to the fact that fewer carbon atoms can be reserved at higher calcination temperatures (Table 1).

Figure 5b compares the CV curves of the NCFC, NCFCO, and NCFO at 10 mV s^−1^. Clearly, all of the CV curves exhibit prominent anodic and cathodic peaks at about −0.7 V and −1.1 V, respectively, which is similar to other Fe-based oxides/carbides [47,48], indicating a dominant Faradic charge storage behavior introduced by the conversion between Fe^2+^ and Fe^3+^, according to the following reaction equations [13,49]:Fe_2_O_3_ + 2e^−^ + 3H_2_O ⇌ 2Fe(OH)_2_ + 2OH^−^(1)
Fe_3_O_4_ + 2e^−^ + 4H_2_O ⇌ 3Fe(OH)_2_ + 2OH^−^(2)

Herein, the Fe_3_O_4_ species should be generated through the electrochemically induced activation of Fe_3_C, during the initial charge–discharge process [14]. Compared to the NCFC and NCFO, the CV curve of the NCFCO exhibits the largest integral area, and prominent redox peaks, proving the highest specific capacitance and rapid redox reaction kinetics, respectively. Additionally, the anodic and cathodic peaks in the CV curves (Appendix A) shift to a more negative and positive region with the increasing scan rate. Such results could be ascribed to the polarization originating from the accumulation of ions with the increment of the scan rate. Furthermore, the apparent anodic and cathodic peaks at the scan rates from 5 to 100 mV s^−1^ can be clearly observed (Appendix A), demonstrating a significant pseudocapacitive behavior in terms of Equations (1) and (2), and the superior capacitive performance of the NCFC, NCFCO, and NCFO. The CV curves of AC (Appendix A) display approximately rectangular shapes at different scan rates, revealing the dominant EDLC behavior.

The GCD profiles at 2 A g^−1^ are compared in Figure 5c. As presented, the nonlinear GCD plots with apparent voltage platforms at around −0.7 and −1.1 V confirm the significant Faradic reactions of the products, which are consistent with their CV results. Additionally, the longest discharge time of the NCFCO reveals its maximum specific capacitance. The GCD plots at diverse current densities display apparent potential plateaus (Appendix A), verifying the excellent electrochemical reversibility and capacitive property. The GCD curves of AC (Appendix A) show nearly-isosceles-triangle-shaped profiles at any current density, which is a typical feature of the EDLC. Figure 5d displays the specific capacitance vs. current density plots of the samples. As seen, the specific capacitance reduces with the increase in the current density, as a result of the active sites being unable to complete the necessary number of redox conversions and ion migrations during the rapid charge–discharge process. Furthermore, the NCFCO delivers the specific capacitance of 590.1 F g^−1^ at 2 A g^−1^, superior to that of the NCFC (261.7 F g^−1^) and NCFO (398.3 F g^−1^), and it remains at 196.7 F g^−1^ at 20 A g^−1^, exhibiting a good rate capability. The comparisons of our NCFCO and the previously reported Fe-based oxides/carbides are listed in Appendix A. Clearly, our NCFCO exhibits an outstanding specific capacitance, which is likely derived from the unique hybrid architecture of the Fe_3_C/Fe_2_O_3_/N-doped ECNF composite.

An asymmetric SC device was assembled using the NCFCO anode and AC cathode, and its electrochemical performance was investigated in a two-electrode system. The CV plots of the NCFCO//AC device at diverse scan rates are recorded in Figure 6a. As revealed, the remarkable redox peaks in the CV curves can be attributed to the Faradic reactions of Fe_2_O_3_ and Fe_3_O_4_ species, according to Equations (1) and (2). Furthermore, no apparent deformation can be observed, even at high scan rates, proving the excellent capacitive performance. In agreement with the CV results, the nonlinear GCD profiles (Figure 6b) of the asymmetric device at different current densities display evident anodic and cathodic platforms in the charging and discharging steps, respectively, which further confirm the existence of Faradic reactions. The specific capacitances of the NCFCO//AC device derived from the GCD results are calculated and plotted in Figure 6c. Specifically, a high specific capacitance of 40 F g^−1^ at 1 A g^−1^ is achieved, and is maintained at 12.5 F g^−1^ (31.3% of the initial value) at 20 A g^−1^. In order to investigate the maximum working voltage window of the as-assembled NCFCO//AC, the CV curves at 10 mV s^−1^ under various voltage windows are recorded and displayed in Figure 6d. Clearly, the stable operating voltage window can extend to 1.9 V without any irreversible oxygen/hydrogen evolution reactions occurring. Generally, the cycling stability is also a crucial performance indicator for an SC. As depicted in Figure 6e, the specific capacitance can still retain 96.7% of the original value after 10,000 cycles at a current density of 10 A g^−1^, and the last ten GCD cycles display negligible change compared to the first ten GCD cycles. Compared to the initial morphology (Appendix A) of the NCFCO electrode, almost no structural destructions can be observed after 5000 (Appendix A) and 10,000 (Appendix A) charge–discharge cycles, which further confirms that the Fe_3_C/Fe_2_O_3_ nanoparticles are tightly anchored on the N–doped ECNFs, to form a robust hybrid architecture. Meanwhile, the corresponding charge efficiency can still maintain 100% after long-cycling, confirming the excellent energy transmission capability. Such results demonstrate the excellent cycling stability and reversibility of the asymmetric NCFCO//AC device.

Figure 7 presents the Ragone plot of the NCFCO//AC device. Clearly, this asymmetric SC exhibits a competitive energy density of 14.2 Wh kg^−1^ at 800 W kg^−1^, and still remains 5 Wh kg^−1^ at 12,000 W kg^−1^, which is higher than some of the Fe-based SCs, such as NiCo_2_S_4_/graphene//Fe_2_O_3_/graphene (5 Wh kg^−1^ at 595 W kg^−1^) [50], NiO//Fe_2_O_3_ (12.4 Wh kg^−1^ at 951 W kg^−1^) [51], Fe_2_O_3_//Fe_2_O_3_ (4.2 Wh kg^−1^ at 224.9 W kg^−1^) [52], NiO@N-doped carbon nanoarrays/carbon cloth//Fe_2_O_3_@N–doped carbon nanoarray/carbon cloth (12.23 Wh kg^−1^ at 307.69 W kg^−1^) [25], and Fe_3_O_4_/N–doped carbon nanosheet/carbon nanotube (11 Wh kg^−1^ at 2500 W kg^−1^) [53]. The outstanding electrochemical energy storage performance of our NCFCO makes it a superior candidate for high-performance SCs.

## 3. Experimental

### 3.1. Chemicals and Regents

N-dimethylformamide (DMF, AR), potassium hydroxide (KOH, AR), polyacrylonitrile (PAN, M_w_ = 150,000, Sigma-Aldrich, Shanghai, China), iron acetylacetonate (IAA, AR), and N-methylpyrrolidone (NMP, AR) were all bought from Sinopharm Chemical Reagent Co., Ltd. (Shanghai, China), and used as received. AC was bought from Jiangsu Xianfeng Nanomaterial Technology Co., Ltd. (Nanjing, China). The deionization water used in this work was fabricated using an ultrapure water system.

### 3.2. Preparations of the NCFC, NCFCO, and NCFO

The schematic illustrations of the preparation procedures for the NCFCO are displayed in Figure 1. Typically, 0.6 g IAA was added into 4.5 g DMF, and stirred for 1 h at room temperature, to prepare a homogeneous solution. After that, 0.5 g PAN was dissolved into the above solution, and stirred at 60 °C for 6 h, to form a uniform spinning solution. Immediately, electrospinning was conducted using an applied voltage of 12 kV, with a flow rate of 0.7 mL h^−1^, at room temperature. Next, the as-obtained nanofiber precursor was collected, and then carbonized at 800 °C (5 °C min^−1^) for 2 h in a N_2_ atmosphere. The as-prepared sample was named as the NCFC. Finally, the NCFC was individually oxidized at 200 °C (5 °C min^−1^) and 400 °C (5 °C min^−1^) in air for 2 h, to fabricate the NCFCO and NCFO, respectively. For comparison, ECNFs were also prepared using a similar procedure to that of the NCFC, except for the addition of IAA. The samples and corresponding preparation conditions are listed in Appendix A.

The electrochemical measurements and characterizations applied in this work are shown in the Supporting Information.

## 4. Conclusions

In summary, a N-doped-ECNF-supported Fe_3_C/Fe_2_O_3_ nanoparticle hybrid architecture composite was successfully synthesized through a facile electrospinning and subsequent thermal treatment strategy. The optimized hybrid architecture, with a large SSA, endows the NCFCO with low charge transfer resistance, and fast redox kinetics. Relying on the synergistic contributions of the Fe_3_C/Fe_2_O_3_ nanoparticles and NCNFs, the as-obtained NCFCO shows the high specific capacitance of 590.1 F g^−1^ at 2 A g^−1^. The asymmetric SC device based on the NCFCO anode and AC cathode can output a high energy density of 14.2 Wh kg^−1^ at 800 W kg^−1^. Furthermore, it also displays long-term cycling stability, with only a 3.3% capacitance decay after 10,000 cycles. Our work demonstrates a facile process to construct Fe-based oxides/carbides/NCNFs for SCs.

## Figures and Tables

**Figure 1 molecules-28-05751-f001:**
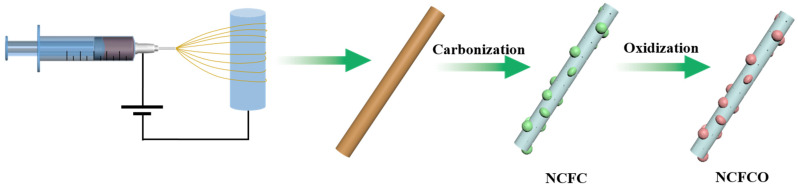
Schematic diagram of the procedure for synthesizing the NCFCO.

**Figure 2 molecules-28-05751-f002:**
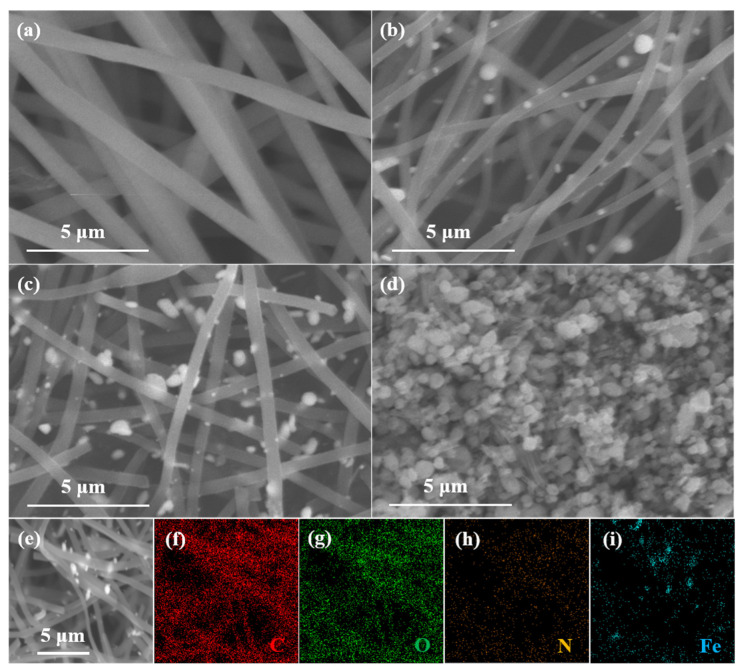
SEM images of the (**a**) ECNF, (**b**) NCFC, (**c**) NCFCO, and (**d**) NCFO. SEM image (**e**) of the NCFCO and the corresponding elemental mappings of (**f**) C, (**g**) O, (**h**) N, and (**i**) Fe, respectively.

**Figure 3 molecules-28-05751-f003:**
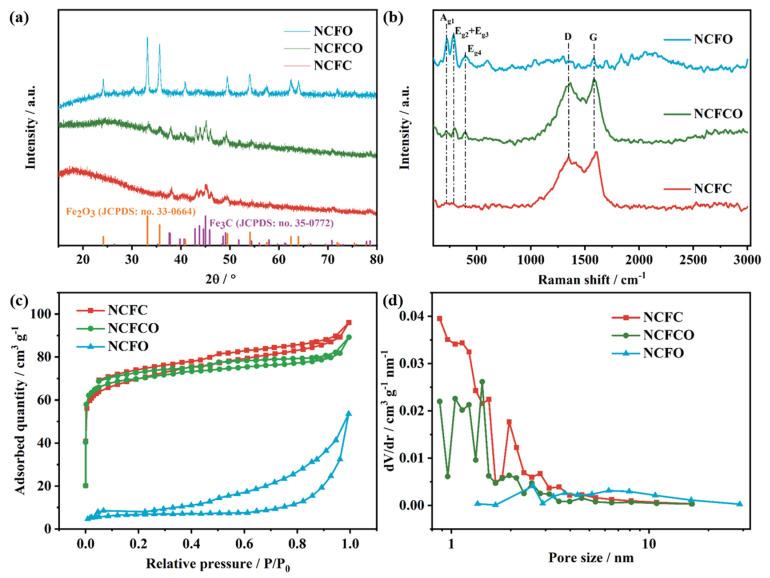
The (**a**) XRD patterns, (**b**) Raman spectroscopy, (**c**) N_2_ adsorption–desorption isotherms, and (**d**) pore size distribution of the NCFC, NCFCO, and NCFO.

**Figure 4 molecules-28-05751-f004:**
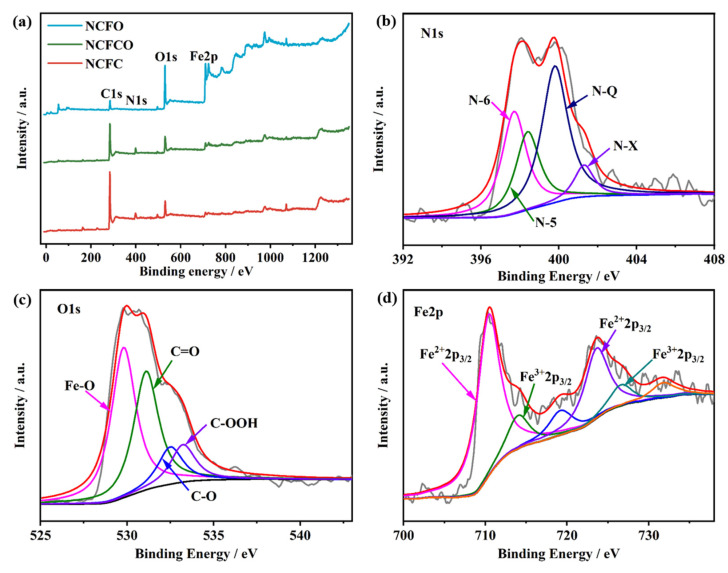
The (**a**) XPS survey spectra of the NCFC, NCFCO, and NCFO. The high-resolution (**b**) C1, (**c**) N1, and (**d**) Fe2p peaks of the NCFCO.

**Figure 5 molecules-28-05751-f005:**
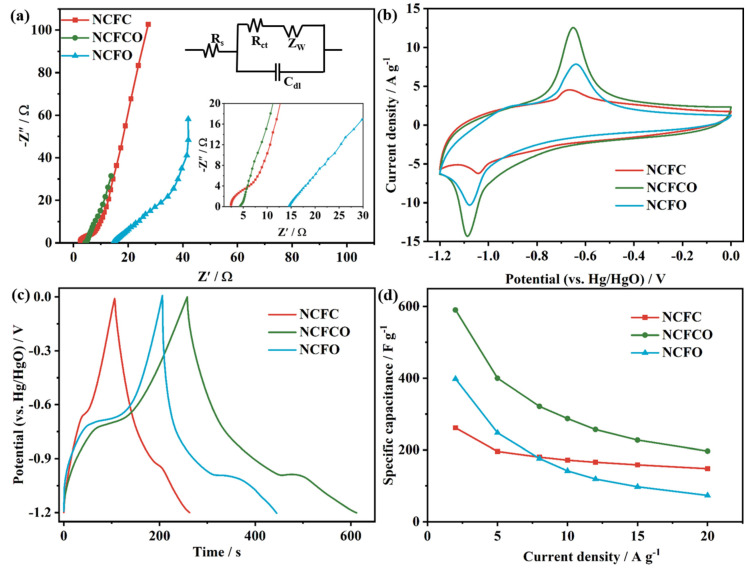
The (**a**) Nyquist spectra (inset shows the high-resolution plots in the low frequency region and the equivalent circuit), (**b**) CV curves at 10 mV s^−1^, (**c**) GCD plots at 2 A g^−1^, and (**d**) specific capacitance vs. current density plots of the NCFC, NCFCO, and NCFO.

**Figure 6 molecules-28-05751-f006:**
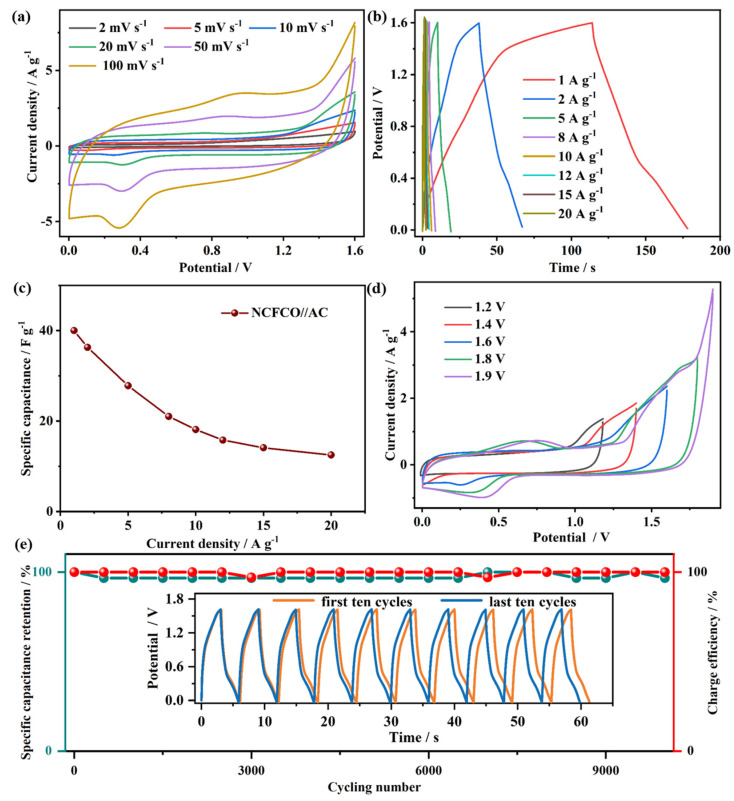
The (**a**) CV curves, (**b**) GCD plots, (**c**) specific capacitance vs. current density plot, (**d**) CV curves at 10 mV s^−1^ with diverse voltage windows, and (**e**) cyclic stability and corresponding charge efficiency (inset shows the first and last ten GCD cycles) of the NCFCO//AC device.

**Figure 7 molecules-28-05751-f007:**
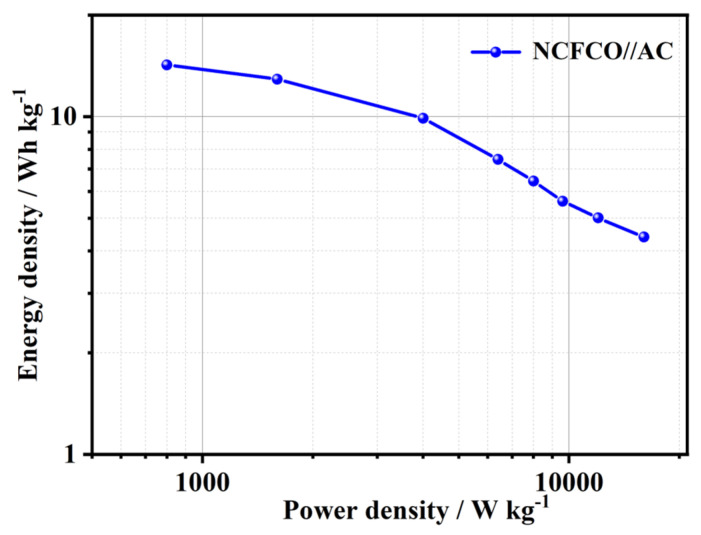
The Ragone plot of the NCFCO//AC device.

**Table 1 molecules-28-05751-t001:** The atomic contents of the NCFC, NCFCO, and NCFO.

	C/%	N/%	O/%	Fe/%
NCFC	79.7	7.6	10.8	1.9
NCFCO	73.1	8.5	15.2	3.2
NCFO	27.8	2.5	46.2	23.5

## Data Availability

Not applicable.

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
