# Peer review of "N-Doped Porous Carbon-Nanofiber-Supported Fe3C/Fe2O3 Nanoparticles as Anode for High-Performance Supercapacitors"

_molecules, 2023, doi:10.3390/molecules28155751_

Round 1

Reviewer 1 Report

The authors synthesized a good supercapacitor and devoted their work to studying its properties. There are a number of comments, none of which can be classified as major, but there are quite a lot of minor comments.

1. Line 59-69, The purpose of the work is formulated as a ready-made result: ” we successfully fabricated Fe2O3/Fe3C anchored N-doped ECNFs (NCFCO) hybrid structure nanocomposite by carbonizing and oxidizing the electrospun nanofiber  precursor containing polyacrylonitrile and iron acetylacetonate. And (what for «and”?) the electrochemical performances were assessed by using different electrochemical measurements. It was  found that the NCFCO exhibited remarkably reinforced specific capacitance of 590.1 F g-1  at 2 A g-1, much higher than those of Fe3C anchored N-doped ECNFs (NCFC, 261.7 F g-1) and Fe2O3 nanoparticles/N-doped ECNFs composite (NCFO, 398.3 F g-1). Importantly, the  asymmetric SC based on our NCFCO anode and activated carbon (AC) cathode delivered  high energy density and exhibited a long-term cycling life. Our work provides a facile  method to integrate Fe-based oxides/carbides with ECNFs to construct high-performance  electrode material for SC applications.”   

2. Patent RU 2 735 854 C9 notes such high qualities of cobalt oxide Co3O4 as pronounced reversibility of redox properties, large surface area, high conductivity, long-term stability and good corrosion stability. What are the advantages of the nanocomposite proposed by the authors in comparison with cobalt-based composites.

3. Section 2. It is recommended to give a table where you can list all the samples and conditions for their preparation, and other features.

4. It is recommended to divide Part 3 into sections describing the relevant measurements.

5. Figure 5.a: not visible scales on the insert. b: the caption does not indicate the composition of the electrolyte in which the CV was obtained.

6. Line 213-214.  It is necessary to indicate the reference from where Equations 1 and 2 are taken. It is necessary to change the type of reversibility arrows to the generally accepted one.

7. Line 210.  The potentials of a pair of peaks are given. In this case, two equations are given. Where are the other pair of peaks for equation (1) or (2)? The process can hardly be called reversible, since the peaks are separated by 400 mV (Figure 5b and Lines 217-218). By a stretch, this is a quasi-reversible process.

8. Figure 6. Caption: a, b, c, d should stand in front of the described content. Figure 6d To which supposed process does the peak belong, starting from the potential of 1 V and further to the anode side? Maybe this is exactly what corresponds to equation 2?

9. List of literature. There is no DOI for all references.

10. Figures S7 and S8. The same questions as in comment 7. In addition, the potential sweep rate of 2 mV/s was used. It is possible for the future work of the authors: it is not recommended to use velocities below 5 mV/s for these CV measurements, since convective mixing of the electrolyte solution begins to contribute (especially since the measurements were carried out under non-thermostated conditions). Although the appearance of curves on the figures is impeccable.

Author Response

To: Katarina ModicEditorMolecules Dear Katarina Modic Manuscript Number: molecules-2463859Title: N-doped porous carbon nanofibers-supported Fe3C/Fe2O3 nanoparticles as anode for high-performance supercapacitors Thank you and reviewers very much for your kind efforts and the constructive comments and suggestions upon which we have revised the manuscript (yellow highlighted) as summarized as follows:

Response to reviewers:

Reviewer: 1

The authors synthesized a good supercapacitor and devoted their work to studying its properties. There are a number of comments, none of which can be classified as major, but there are quite a lot of minor comments.

Reply: Thanks a lot for your kind evaluation and suggestion.

  1. Line 59-69,The purpose of the work is formulated as a ready-made result: ”we successfully fabricated Fe2O3/Fe3C anchored N-doped ECNFs (NCFCO) hybrid structure nanocomposite by carbonizing and oxidizing the electrospun nanofiber precursor containing polyacrylonitrile and iron acetylacetonate. And (what for «and”?) the electrochemical performances were assessed by using different electrochemical measurements. It was found that the NCFCO exhibited remarkably reinforced specific capacitance of 590.1 F g-1 at 2 A g-1, much higher than those of Fe3C anchored N-doped ECNFs (NCFC, 261.7 F g-1) and Fe2O3 nanoparticles/N-doped ECNFs composite (NCFO, 398.3 F g-1). Importantly, the asymmetric SC based on our NCFCO anode and activated carbon (AC) cathode delivered high energy density and exhibited a long-term cycling life. Our work provides a facile method to integrate Fe-based oxides/carbides with ECNFs to construct high-performance  electrode material for SC applications.”

Reply 1-1: Thanks a lot for your valuable comment. Relative statements have been revised. Please see Line 57-72 and Line 75-79 in the revised manuscript.

  1. Patent RU 2 735 854 C9 notes such high qualities of cobalt oxide Co3O4 as pronounced reversibility of redox properties, large surface area, high conductivity, long-term stability and good corrosion stability. What are the advantages of the nanocomposite proposed by the authors in comparison with cobalt-based composites.

Reply 1-2: Thanks a lot for your valuable comment. Actually, the Co3O4 is an excellent electrode material for supercapacitor and better to be used as cathode, while the Fe-based oxides/carbides are superior anode. Additionally, compared to Co3O4, the Fe-based oxides/carbides exhibit large voltage windows, more efficient cost, and lower toxicity. Furthermore, the NCFCO reported in this work also exhibits large SSA and excellent electrochemical performance. All of these make NCFCO to be an ideal anode for supercapacitors. Please see Line 18-20 in the revised manuscript.

  1. Section 2. It is recommended to give a table where you can list all the samples and conditions for their preparation, and other features.

Reply 1-3: Thanks a lot for your valuable comment. The preparation conditions of the samples are illustrated in Table S1. Please see Line 98-99 in the revised manuscript, Table S1 in the Supporting Information.

  1. It is recommended to divide Part 3 into sections describing the relevant measurements.

Reply 1-4: Thanks a lot for your valuable comment. The Part 3 has been modified as recommended. Please see Line 116 and Line 209 in the revised manuscript.

  1. Figure 5.a: not visible scales on the insert. b: the caption does not indicate the composition of the electrolyte in which the CV was obtained.

Reply 1-5: Thanks a lot for your valuable comment. The scales of the Fig. shown in the inset of Fig. 5(a) have been enlarged. The electrolyte used in the three-electrode system has been illustrated. Please see Line 210, Line 214-217 in the revised manuscript.

  1. Line 213-214. It is necessarytoindicate the reference from where Equations 1 and 2 are taken. It is necessary to change the type of reversibility arrows to the generally accepted one.

Reply 1-6: Thanks a lot for your valuable comment. Relative references have been cited and the arrows in Equations (1) and (2) have been modified. Please see Line 231-233 in the revised manuscript.

  1. Line 210. The potentials of a pair of peaks are given. In this case, two equations are given. Where are the other pair of peaks for equation (1) or (2)? The process can hardly be called reversible, since the peaks are separated by 400 mV (Figure 5b and Lines 217-218). By a stretch, this is a quasi-reversible process.

Reply 1-7: Thanks a lot for your valuable comment. Similar to other Fe-based oxides/carbides (such as DOI: 10.1016/j.jpowsour.2020.228915, 10.1039/d1qi00864a), the redox peaks of Fe2O3 and Fe3O4 in KOH electrolyte generally overlap together, which can not be clearly distinguished and should be due to the approximate redox potential between Fe2+ and Fe3+ of Fe2O3 and Fe3O4. Relative statements have been modified. Please see Line 227-231, Line 235, and Line 241-244 in the revised manuscript.

  1. Figure 6. Caption: a, b, c, d should stand in front of the described content. Figure 6d To which supposed process does the peak belong, starting from the potential of 1 V and further to the anode side? Maybe this is exactly what corresponds to equation 2?

Reply 1-8: Thanks a lot for your valuable comment. The captions of Fig. 3, 4, 6, S1, S3, S4, S5, S6, S7, and S9 have been modified. Because of the approximate redox potential between Fe2+ and Fe3+ of Fe2O3 and Fe3O4 species, in Fig. 6(d), the redox peaks in the CV curves maybe caused by the co-action of Fe2O3 and Fe3O4 species by occurring Faradic reactions. Please see Line 142-144, Line 178-180, Line 263-266, and Line 269-271 in the revised manuscript, Fig. S1, S3, S4, S5, S6, S7, and S9 in the Supporting Information.

  1. List of literature. There is no DOI for all references.

Reply 1-9: Thanks a lot for your valuable comment. The format of references shown in the manuscript have been edited according to the requirements of the journal of Molecules.

  1. Figures S7 and S8. The same questions as in comment 7. In addition, the potential sweep rate of 2 mV/s was used. It is possible for the future work of the authors: it is not recommended to use velocities below 5 mV/s for these CV measurements, since convective mixing of the electrolyte solution begins to contribute (especially since the measurements were carried out under non-thermostated conditions). Although the appearance of curves on the figures is impeccable.

Reply 1-10: Thanks a lot for your valuable comment. Similar to other reports (such as DOI: 10.1016/j.jpowsour.2020.228915, 10.1039/d1qi00864a), the redox peaks and charge-discharge platforms of Fe2O3 and Fe3O4 in KOH electrolyte generally overlap together, which can not be clearly distinguished and should be due to the approximate redox potential between Fe2+ and Fe3+ of Fe2O3 and Fe3O4 species. The CV curves of the samples have been modified. Please see Line 241-244 in the revised manuscript, Fig. S7 and S8 in the Supporting Information.

Again, we sincerely appreciate the editor and reviewers’ comments. We hope that the revision has fully addressed the issues brought up in the review.

Sincerely yours,

Pinghua Zhang

Key Laboratory of Spin Electron and Nanomaterials of Anhui Higher Education Institutes, Suzhou University, SuZhou 234000, PR China.

[email protected]

Reviewer 2 Report

The authors synthesized a wire-dot hybrid type capacitor. And interestingly it has shown recyclable nature up to 10000 cycles.

The resulting materials have been characterized through elemental analysis, energy band analysis, and the key electrochemical properties were proved.

However the recycled nanoparticles normally show deformation and the surface to volume ratio decreases during the recycle process. And this induces deteriorated surface properties, here the electrochemical properties. Therefore the authors need to suggest SEM micrographs depending on cycle numbers as additional key data.

Author Response

To: Katarina ModicEditorMolecules Dear Katarina Modic Manuscript Number: molecules-2463859Title: N-doped porous carbon nanofibers-supported Fe3C/Fe2O3 nanoparticles as anode for high-performance supercapacitors Thank you and reviewers very much for your kind efforts and the constructive comments and suggestions upon which we have revised the manuscript (yellow highlighted) as summarized as follows:

Response to reviewers:

Reviewer: 2

Comments and Suggestions for Authors

The authors synthesized a wire-dot hybrid type capacitor. And interestingly it has shown recyclable nature up to 10000 cycles. The resulting materials have been characterized through elemental analysis, energy band analysis, and the key electrochemical properties were proved.

Reply: Thanks a lot for your kind evaluation and suggestion.

  1. However the recycled nanoparticles normally show deformation and the surface to volume ratio decreases during the recycle process. And this induces deteriorated surface properties, here the electrochemical properties. Therefore the authors need to suggest SEM micrographs depending on cycle numbers as additional key data.

Reply 2-1: Thanks a lot for your valuable comment. The SEM images of NCFCO electrode after different cycling numbers are shown in Fig. S11. Please see Line 286-290 in the revised manuscript, Fig. S11 in the Supporting Information.

Again, we sincerely appreciate the editor and reviewers’ comments. We hope that the revision has fully addressed the issues brought up in the review.

Sincerely yours,

Pinghua Zhang

Key Laboratory of Spin Electron and Nanomaterials of Anhui Higher Education Institutes, Suzhou University, SuZhou 234000, PR China.

[email protected]

Round 2

Reviewer 2 Report

The revised form reflecs the review comments.